# Is minor surgery safe during the COVID-19 pandemic? A multi-disciplinary study

**Michael Baboudjian** [1]*, **Mehdi Mhatli**[2,3], **Adel Bourouina**[4], **Bastien Gondran-Tellier**[1], **Vassili Anastay**[1], **Lea Perez**[1], **Pauline Proye**[1], **Jean-Pierre Lavieille**[3], **Fanny Duchateau**[5], **Aubert Agostini**[5], **Yann Wazne**[4], **Frederic Sebag**[4], **Jean-Marc Foletti**[6], **Cyrille Chossegros**[6], **Didier Raoult**[7], **Julian Touati** [8], **Christophe Chagnaud**[8], **Justin Michel**[2], **Baptiste Bertrand**[9], **Antoine Giovanni**[2], **Thomas Radulesco**[2], **Catherine Sartor**[10], **Pierre-Edouard Fournier**[7], **Eric Lechevallier**[1]

1 Department of Urology and Kidney Transplantation, Aix-Marseille University, APHM, Conception University Hospital, Marseilles, France, 2 Department of Otorhinolaryngology- Head & Neck Surgery, APHM, Aix-Marseille University, Conception University Hospital, Marseilles, France, 3 Department of Head and Neck Surgery, Conception University Hospital, Aix-Marseille University, Marseilles, France, 4 Department of Endocrine Surgery, Conception University Hospital, Aix-Marseille University, Marseilles, France, 5 Department of Obstetrics and Gynecology, Conception Hospital, Marseilles, France, 6 Department of Oral and maxillofacial Surgery, Aix Marseille University, APHM, IFSTTAR, LBA, Conception University Hospital, Marseilles, France, 7 Aix-Marseille University, IRD, AP-HM, IHU Méditerranée Infectious Disease Research Institute, Marseilles, France, 8 Department of Radiology, Conception University Hospital, APHM, Marseilles, France, 9 Department of Plastic Surgery, Conception University Hospital, APHM, Marseilles, France, 10 Operational Hospital Hygiene Team, Conception University Hospital, Marseilles, France

* Michael.BABOUDJIAN@ap-hm.fr

**Data Availability Statement:** All relevant data are within the manuscript and its Supporting Information files.

## Abstract

### Background

To assess the risk of postoperative SARS-CoV-2 infection during the COVID-19 pandemic.

### Methods

The CONCEPTION study was a cohort, multidisciplinary study conducted at Conception University Hospital, in France, from March 17th to May 11th, 2020. Our study included all adult patients who underwent minor surgery in one of the seven surgical departments of our hospital: urology, digestive, plastic, gynecological, otolaryngology, gynecology or maxillofacial surgery. Preoperative self-isolation, clinical assessment using a standardized questionnaire, physical examination, nasopharyngeal RT-PCR and chest CT scan performed the day before surgery were part of our active prevention strategy. The main outcome was the occurrence of a SARS-CoV-2 infection within 21 days following surgery. The COVID-19 status of patients after discharge was updated during the postoperative consultation and to ensure the accuracy of data, all patients were contacted again by telephone.

### Results

A total of 551 patients from six different specialized surgical Departments in our tertiary care center were enrolled in our study. More than 99% (546/551) of included patients underwent a complete preoperative Covid-19 screening including RT-PCR testing and chest CT scan

**Funding:** The authors received no specific funding for this work.

**Competing interests:** The authors have declared that no competing interests exist.

upon admission to the Hospital. All RT-PCR tests were negative and in 12 cases (2.2%), preoperative chest CT scans detected pulmonary lesions consistent with the diagnosis criteria for COVID-19. No scheduled surgery was postponed. One patient (0.2%) developed a SARS-CoV-2 infection 20 days after a renal transplantation. No readmission or COVID-19 -related death within 30 days from surgery was recorded.

## Conclusions

Minor surgery remained safe in the COVID-19 Era, as long as all appropriate protective measures were implemented. These data could be useful to public Health Authorities in order to improve surgical patient flow during a pandemic.

## Introduction

From December 2019 onwards, severe acute respiratory syndrome coronavirus 2 (SARS-CoV-2) has quickly spread worldwide, leading to a global pandemic, and drastically altering everyday life and clinical practices [1]. Adequate testing, early diagnosis, isolation and contact tracing have proved to be key measures to control the spread of SARS-CoV-2 [2]. Nasopharyngeal reverse transcriptase polymerase chain reaction (RT-PCR) is currently the most commonly used method for diagnosing coronavirus disease 2019 (COVID-19) [3]. In order to offset the higher false negative rate for SARS-CoV-2 RT- PCR, chest Computed Tomography (CT) scan has been proposed as an interesting option for COVID-19 screening and early diagnosis [4, 5]. Based on current evidence, CT scans have been shown to have a higher sensitivity earlier in the time course of infection than conventional RT-PCR in symptomatic patients. However, their utility is still being debated in asymptomatic patients, especially regarding preoperative surgical patients [6].

Public Health authorities recommend reducing surgical activities during pandemic peaks. The COVID-19 pandemic has forced surgical departments to re-schedule their activity, giving priority to urgent procedures and non-deferrable oncological cases [7]. Only a very few studies have focused on the collateral effects of the COVID-19 pandemic [8–10]. Furthermore, delaying non-urgent surgical procedures has resulted in a significant backlog in elective surgeries. This has not only impacted the capacity of the surgical system but also aroused in many patients the fear of contracting COVID-19 in hospital [11].

Since the first peak of the COVID-19 pandemic in France, in joint agreement with the *Méditerranée Infectious Disease Research Institute (IHU)*, we have implemented rigorous preoperative screening as part of a set of active preventive measures including a systematic RT-PCR test and a chest CT scan performed the day before surgery. The aim of the present study is to assess the postoperative SARS-CoV-2 infection risk when systematic, active, preventive measures are duly applied.

## Materials & method

### Study population

The study was submitted to the Ethics Committee of the French Society for Urology. After obtaining Institutional Review Board approval (RGPD/ Assistance Publique des Hôpitaux de Marseille, Ap-Hm, n˚2020–295), we retrospectively identified all adult patients who underwent mainly minor surgery at Conception University hospital between March 17th and May

11th, 2020. All of the seven surgical departments in our hospital were involved in our study: urology, digestive, plastic, otolaryngology (two departments), gynecology or maxillofacial surgery. No exclusion criteria were applied, as long as patients were operated on during the study period.

The following baseline characteristics were collected from the patients' electronic medical records: patient age and gender, Body Mass Index (BMI), comorbidities, age-adjusted Charlson comorbidity index, residence (home or institution), date of surgery, type of surgery, emergency or scheduled indications, type of anesthesia (general, spinal, local), preoperative SARS-CoV-2 RT-PCR test result, preoperative chest CT scan result, and length of hospital stay.

## Perioperative prevention of Covid-19

Preoperative self-isolation at least 7 days before admission was systematically recommended. During the study period, patients were hospitalized in single-patient rooms as much as possible. External visitors were not allowed. Disabled patients were allowed only one support person during their hospital stay. Strict conditions were imposed: a negative RT-PCR test was required, and the support person should remain in the patient's room.

Verbal informed consent was obtained from all patients before preoperative routine SARS-CoV-2 screening, and patients were aware that these data would be used anonymously for research purposes. In joint agreement with the nosocomial infection control committee and the *Méditerranée Infectious Disease Research Institute* (IHU), our hospital facility implemented preoperative screening as part of the recommended active preventive measures for COVID-19 during the peak of the pandemic. These measures mainly consisted of social distancing [12], frequent handwashing and disinfection, and the use of personal protective equipment such [13] as gloves, goggles, face shields and masks. All patients were systematically screened for COVID-19 symptoms prior to surgery, on the basis of a standardized questionnaire. This questionnaire was designed in an emergency situation and validated by the local crisis committee of the *Assistance Publique des Hôpitaux de Marseille*. It enabled us to look out for previously reported symptoms of COVID-19 such as fever, flu-like syndrome, anosmia, ageusia or digestive symptoms, as well as, for patients who might have been a close contact of someone with COVID-19 (Fig 1). Testing of nasopharyngeal swab specimens by RT-PCR was systematically performed the day before surgery [14]. Then, all patients were systematically subjected to chest CT scan before surgery. In order to be able to perform preoperative screening in each patient, specific time- slots were reserved by the imaging department the day before surgery. Radiographers were trained to acquire images while minimizing pathogen exposure to staff. In a standardized report, two radiographers (with 4 and 30 years of experience, respectively) devised a classification of images into four grades: no abnormalities on Chest CT scan, low, moderate, or high suspicion for COVID-19. On this basis, a multi-disciplinary scientific advisory board was set up so as to be able to provide personalized care to patients.

## Follow-up and endpoint

The patients were operated on in one of the hospital's seven surgical departments: urology, digestive, plastic, gynecological, otolaryngology, gynecology or maxillofacial surgery. In our hospital, there was no cardiac, vascular, neurological or orthopedic surgery. The primary endpoint of our study was the occurrence of SARS-CoV-2 infection within 21 days postoperatively. Postoperative assessment of COVID-19 symptoms was performed systematically during hospitalization. Patients were discharged and instructed to return to the hospital so as to get retested for COVID-19 with RT-PCR if any specific symptom developed in the meantime. In

**Have you been diagnosed with COVID?**

Yes ☐, date……… No ☐

**In previous days, did you present with?**

| | | | |
|---|---|---|---|
| ➢ | Fever | Yes ☐ | No ☐ |
| ➢ | Tiredness | Yes ☐ | No ☐ |
| ➢ | Dry cough | Yes ☐ | No ☐ |
| ➢ | Rhinorrhea | Yes ☐ | No ☐ |
| ➢ | Shortness of breath | Yes ☐ | No ☐ |
| ➢ | Chest pain | Yes ☐ | No ☐ |
| ➢ | Body aches | Yes ☐ | No ☐ |
| ➢ | Loss of smell | Yes ☐ | No ☐ |
| ➢ | Loss of taste | Yes ☐ | No ☐ |
| ➢ | Diarrhea | Yes ☐ | No ☐ |
| ➢ | Headache | Yes ☐ | No ☐ |

If yes, symptom onset date: ………………………..

➢ None of these symptoms            ☐

**Have you been in close contact with a symptomatic person?**

Yes ☐, date……… No ☐

**Consent for COVID-19- Diagnostic Testing? Yes ☐ No ☐**

**Fig 1. Preoperative COVID-19 symptom-based screening questionnaire.**

order to update the COVID-19 status of discharged patients all of them were assessed during the postoperative consultation which could be either a telemedicine consultation or a face-to-face consultation. In order to ensure the accuracy of our data, all patients were phoned again in December 2020. A patient was considered a confirmed COVID-19 case if he/she had a positive RT-PCR test result or developed specific pulmonary lesions visible on a chest CT scan performed after surgery.

## Data analysis

Demographic data, preoperative clinical information, perioperative and follow-up variables were extracted from E-medical records and recorded in a dedicated database. Descriptive statistics were conducted on available variables. Quantitative variables were reported as medians and interquartile ranges [IQR]. Categorical variables were described by numbers and percentages. All statistical analyses were performed using R statistical software Version 4.0.2. (Foundation for Statistical Computing, Vienna, Austria).

## Results

Between March 17 and May 11, 2020, 551 surgical patients from six different specialized surgical Departments in our University Hospital were included in our study. Most hospitalized patients had been advised to stay at home prior to surgery (n = 513, 93.1%). 38 patients (6.9%) resided in a nursing home or a rehabilitation center.

Patient baseline characteristics are summarized in Table 1. In the whole cohort, the median age was 63 (IQR 51–71) years, 254 female patients (46.1%) were included, the median BMI was 24.6 (IQR 21.7–28), and the median Charlson Comorbidity score was 3 (IQR 1–4). The main comorbidities were: circulatory diseases (n = 212, 38.5%), arterial hypertension (n = 200,

**Table 1. Baseline characteristics upon admission.**

|  | Overall cohort (n = 551) |
|---|---|
| **Gender, *n (%)*** |  |
| Male | 297 (53.9) |
| Female | 254 (46.1) |
| **Median (IQR) age, *years*** | 63 (51–71) |
| **Median (IQR) Body mass index** | 24.6 (21.7–28) |
| **Median (IQR) age-adjusted Charlson Comorbidity Index** | 3 (1–4) |
| **Residence, *n (%)*** |  |
| Home | 513 (93.1) |
| Institution | 38 (6.9) |
| **Comorbidity, *n (%)*** |  |
| Cardiovascular | 212 (38.5) |
| Pulmonary | 76 (13.8) |
| Hepatic | 26 (4.7) |
| Neurological | 50 (9) |
| Diabetes mellitus | 82 (14.9) |
| Arterial High Blood Pressure | 200 (36.3) |
| Immunosuppression | 56 (10.2) |
| **Type of surgery, *n (%)*** |  |
| Acute Surgical Emergencies | 28 (5.1) |
| Elective surgery | 523 (94.9) |
| **Cancer surgery, *n (%)*** |  |
| Yes | 276 (50.1) |
| No | 275 (49.9) |
| **Renal transplantation, *n (%)*** | 15 (2.7) |
| **Type of anesthesia, *n (%)*** |  |
| General | 524 (95.1) |
| Spinal | 12 (2.2) |
| Local | 15 (2.7) |

**Table 2. Preoperative screening test results.**

|  | Overall cohort (n = 551) |
|---|---|
| **RT-PCR, *n (%)*** |  |
| Negative | 546 (99.1) |
| Positive | 0 (0) |
| Not performed | 5 (0.9) |
| **CT chest scan, *n (%)*** |  |
| Normal | 537 (97.5) |
| Low suspicion for COVID-19 | 9 (1.6) |
| Moderate suspicion for COVID-19 | 3 (0.5) |
| High suspicion for COVID-19 | 0 (0) |
| Not performed | 2 (0.4) |
| **Incidentaloma, *n (%)*** |  |
| No | 460 (89.1) |
| Yes | 60 (10.9) |

RT-PCR: Reverse transcriptase polymerase chain reaction; CT: Computed Tomography.

36.3%), and respiratory diseases (n = 76, 13.8%). Regarding the types of surgery, 523/551 (94.9%) patients had an elective surgery and 276 (50.1%) had oncological surgery. General anesthesia was the most commonly used type of anesthesia (n = 524, 95.1%).

As shown in Table 2, more than 99% of included patients had a complete preoperative screening including RT-PCR test and chest CT scan upon admission. Seven patients only had not been subjected to RT-PCR tests and chest CT scans before surgery. All RT-PCR tests proved negative. In 537 cases (97.5%), preoperative chest CT scan showed no obvious abnormalities. Conversely, it showed that twelve patients with negative RT-PCR tests had low or moderate suspicion of COVID-19. All of them benefitted from multi-disciplinary personalized care, none of them had their surgery postponed. In 60 patients, incidental findings were diagnosed. In total, 13 lung lesions suspect of malignancy were found. A total of 32 pulmonary nodules requiring follow-up were reported. Lastly, interstitial lung disease (n = 5), aortic aneurysm (n = 4), pleural effusion (n = 4), thyroid goiter (n = 1) and dilatation of the renal cavities (n = 1) were fortuitously discovered. Any incidental finding was reported to the general practitioner.

Surgical outcomes are shown in Table 3. The median length of stay was 3 days (IQR 2–4). Due to the closure of the outpatient surgery department during the study period, only six patients were treated as outpatients and were discharged from hospital the same day. A

**Table 3. Postoperative outcomes.**

|  | Overall cohort (n = 551) |
|---|---|
| **Median (IQR) length of stay, *days*** | 3 (2–4) |
| **21-d infection with SARS-CoV-2, *n (%)*** | 1 (0.2) |
| **30-d readmission, *n (%)*** |  |
| Non-COVID-19-related | 17 (3.1) |
| COVID-19 related | 0 (0) |
| **30-d mortality, *n (%)*** |  |
| Non-COVID-19-related | 0 (0) |
| COVID-19 related | 0 (0) |

comprehensive 30 day- patient follow-up was performed. One patient developed symptoms suggestive of COVID-19 postoperatively, repeated PCR testing turned positive. This patient was hospitalized for a renal transplantation in our nephrology department which is located in another building distinct from the building where our surgical departments are located. He was one of the 2 patients on whom a chest CT scan had not been performed. The immunosuppressed patient developed nosocomial COVID-19 within twenty days of the procedure and required intubation and invasive mechanical ventilation in the intensive care unit. Finally, no readmission or COVID-19-related death was recorded within 30 days of surgery.

## Discussion

Our results have shown that surgery remained safe in COVID-19 Era, as long as all recommended precautions were applied. Our study period corresponds to the first epidemic peak of COVID-19 in France, when the highest number of COVID-19 -related deaths was recorded [12]. Our University Hospital employs 3142 healthcare workers and has a 862- bed -capacity. During the COVID-19 outbreak in the Provence Alpes Côte d'Azur region in France, our hospital was actively involved in the governmental national action plan for confronting the COVID-19 pandemic [15]. Ten additional intensive care beds and 50 hospital beds (distributed in four units) were dedicated to COVID-19 patients. Government provisions promoted outpatient surgery in order to limit the risk of perioperative infection. However, due to a staff shortage in COVID-19 dedicated units, outpatient caregivers were requisitioned and the outpatient surgery department was temporarily closed. Although the estimated median incubation period of COVID-19 is approximately 5 days, delayed symptoms might appear, up to 21 days after exposure to the virus [16]. Therefore, the chosen primary endpoint of our study was a 21 day -period after surgery. The only patient who developed SARS-CoV-2 infection developed symptoms 20 days after receiving a kidney transplant. He was hospitalized in a separate building distinct from the building where our surgical departments are located and thus, no cluster was declared in our care units.

The risk of nosocomial COVID-19 infection has become a sad reality and hospital management has had to adjust day -to—day by issuing guidelines based on COVID-19 pandemic knowledge and behaviors [17]. Various studies have demonstrated that elderly patients admitted in intensive care units are more likely to acquire nosocomial COVID-19 [18, 19]. However, the risks of postoperative SARS-CoV-2 infection have been scarcely evaluated. Our study conducted at Conception University Hospital reflects the efforts of an entire hospital to protect patients and healthcare professionals during the initial phase of the pandemic outbreak in France in March 2020. Our results highlight the value of a rigorous preoperative screening as part of a set of active preventive measures against nosocomial transmission of COVID-19. Healthcare professionals' compliance with the above mentioned preoperative COVID -19 screening protocol was high since more than 99% of included patients had a RT-PCR test and underwent a chest CT scan the day before surgery. Our postoperative COVID-19 infection rate of 0.2% compares favorably with previously reported rates ranging from 1 to 7% [20–23]. Furthermore, no postoperative COVID-19 -related death was reported. Compared with previous studies, the main strength of our study lies in routine use of RT-PCR tests in order to screen patients on admission the day before surgery. It is interesting to note that the high risk of false negative RT-PCR test results at initial testing for COVID-19 [24] apparently had no impact on our results. The medical staff members of the Méditerranée Infection Research Institute are well trained in performing RT-PCR tests, and this may have influenced our results.

The originality of the present study also lies in the use of preoperative chest CT scan as a screening tool for COVID -19 in asymptomatic surgical patients. Out of the 549 patients who had a preoperative chest CT scan, 12 asymptomatic cases were reported. These patients tested negative through RT-PCR and had an abnormal scan (Table 2). All procedures were performed as scheduled and no patient developed any symptoms of COVID-19 postoperatively. Chest CT scan is currently considered as a relevant COVID-19 screening tool in surgical patients who are symptomatic and tested negative for SARS-COV-2 by conventional RT-PCR [6]. However, for asymptomatic surgical patients, no clear medical benefit has been shown in the use of routine scanning of preoperative patients in our case series. According to a previous study [25], in this context, in a small proportion of cases, CT scan could give false- positives resulting in unnecessary delay in surgery, and therefore should not be recommended as a COVID-19 screening tool in asymptomatic surgical patients.

The limitations of the present study should also be acknowledged. Its main limitation lies in its retrospective design. Then, the absence of a control group with similar baseline characteristics including surgical patients screened without a RT-PCR test or a chest CT scan prevents us from drawing any firm conclusions on the relative merits of these COVID-19 screening tools in surgical patients. Furthermore, due to a lack of centralized review of initial CT scans, there remains a risk of inter- and intra-reader variability in chest CT scan assessment, despite using a standardized classification of CT scan results. Finally, the present study reports on a very heterogeneous surgical population and covers a wide range of planned elective surgeries and a variety of emergency surgical presentations. However, our study population consisted mainly of patients who underwent minor elective surgery. This observation should be related to a significant decrease in acute surgical emergencies as observed in previous studies found in the literature [26]. In addition, major surgeries such as cardiac, vascular or neurological surgeries requiring postoperative resuscitation care were not included in our series. Nevertheless, our results are likely to be generalizable to other hospitals and constitute a comprehensive set of data on which a future preoperative screening protocol for COVID-19 could be based.

Our study also has several strengths and could be instrumental in helping clinicians and public health authorities detect COVID-19 in surgical patients. Preoperative self-isolation, clinical assessment using a standardized questionnaire, and RT-PCR tests the day before surgery seem to be effective in minimizing COVID-19- related surgical risk. To update the COVID-19 status of discharged patients, all patients were assessed during the postoperative consultation and in order to ensure the high accuracy of our data, all patients were phoned again in December 2020. The postponement of elective procedures during the peaks of the pandemic was justified by a lack of staff (due to the redeployment of medical staff members to COVID-19 units) and the risk of SARS-CoV-2 infection during hospitalization. Our study has shown that hospital stay during a pandemic peak can remain safe, and therefore only a limited number of surgeries might have to be postponed if additional medical professionals were recruited.

## Conclusion

Minor surgery remained safe in the COVID-19 Era, as long as all recommended precautions were applied. Our postoperative SARS-CoV-2 infection rate was extremely low thanks to a rigorous COVID-19 screening in asymptomatic patients in the preoperative work-up. Preoperative chest CT scan has not been shown to provide any additional benefit in routine preoperative screening of asymptomatic patients. These data may be useful to public health authorities worldwide, in order to improve surgical patient flow during a pandemic.

## Supporting information

**S1 Data.**
(XLSX)

## Acknowledgments

We would like to express our gratitude towards Manon Liot, Alexandre Fournier, Pierre Gambino, Astrid Lefebvre, Louise Lecharny, and Jennifer Lanoe for contacting all patients so as to update their COVID-19 status postoperatively.

## Author Contributions

**Conceptualization:** Michael Baboudjian, Aubert Agostini, Jean-Marc Foletti, Julian Touati, Thomas Radulesco, Pierre-Edouard Fournier, Eric Lechevallier.

**Data curation:** Mehdi Mhatli, Adel Bourouina, Vassili Anastay, Lea Perez, Pauline Proye, Fanny Duchateau, Yann Wazne.

**Formal analysis:** Michael Baboudjian, Bastien Gondran-Tellier.

**Investigation:** Michael Baboudjian.

**Methodology:** Michael Baboudjian, Bastien Gondran-Tellier, Jean-Marc Foletti, Julian Touati, Thomas Radulesco, Eric Lechevallier.

**Software:** Eric Lechevallier.

**Supervision:** Jean-Pierre Lavieille, Aubert Agostini, Frederic Sebag, Cyrille Chossegros, Didier Raoult, Julian Touati, Christophe Chagnaud, Justin Michel, Baptiste Bertrand, Antoine Giovanni, Thomas Radulesco, Catherine Sartor, Pierre-Edouard Fournier, Eric Lechevallier.

**Validation:** Aubert Agostini, Thomas Radulesco, Pierre-Edouard Fournier, Eric Lechevallier.

**Visualization:** Didier Raoult, Christophe Chagnaud, Baptiste Bertrand, Antoine Giovanni, Thomas Radulesco, Pierre-Edouard Fournier.

**Writing – original draft:** Michael Baboudjian, Jean-Marc Foletti, Pierre-Edouard Fournier.

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
