## [Editor Report · Decision Letter 0]

26 Mar 2021

PONE-D-21-08635

Is Surgery Safe During the COVID-19 Pandemic? A Multi-Disciplinary Study

PLOS ONE

Dear Dr. Baboudjian,

Thank you for submitting your manuscript to PLOS ONE. After careful consideration, we feel that it has merit but does not fully meet PLOS ONE’s publication criteria as it currently stands. Therefore, we invite you to submit a revised version of the manuscript that addresses the points raised during the review process.

Please specify "surgery" upfront in the Title and also in Methods.  Did it include open heart surgery, TAVR, TEVAR, aortic dissection, brain surgery, dental surgery, Whipple operation, Bentall operation, .VATS,...too vague and confusing!

We look forward to receiving your revised manuscript.

Kind regards,

Academic Editor

PLOS ONE

Journal Requirements:

1. Please ensure that your manuscript meets PLOS ONE's style requirements, including those for file naming. The PLOS ONE style templates can be found athttps://journals.plos.org/plosone/s/file?id=wjVg/PLOSOne_formatting_sample_main_body.pdf andhttps://journals.plos.org/plosone/s/file?id=ba62/PLOSOne_formatting_sample_title_authors_affiliations.pdf

3. Thank you for providing the date(s) when patient medical information was initially recorded. Please also include the date(s) on which your research team accessed the databases/records to obtain the retrospective data used in your study.

4. Thank you for stating in the text of your manuscript "Verbal informed consent was obtained from all patients before preoperative routine SARSCoV-2 screening, and patients were aware that these data would be used anonymously for research purposes". please describe how verbal consent was documented and witnessed, and why written consent was not obtained. Please also also add all of this information to your ethics statement in the online submission form.

5. Please provide the full name of Institutional Review Board that approved your study.

7. Please include your tables as part of your main manuscript and remove the individual files. Please note that supplementary tables (should remain/ be uploaded) as separate "supporting information" files

8.  Thank you for stating the following financial disclosure:

'The funders had no role in study design, data collection and analysis, decision to

publish, or preparation of the manuscript.'

Additional Editor Comments (if provided):

The scope of "surgery" is ill-defined. Did it include open heart surgery, brain surgery, and some other major surgery? Or just urology surgery? Please specify upfront in the Title. in the Methods, the IR

B is only approved in urology, not other surgical specialty. Was it legal to include non-urology surgery cases? Please specify the scope of "surgery" in the Methods. Open heart surgery? TAVR? TEVAR? EVAR? Brain surgery, Whipple surgery, Bentall surgery, type-A aortic dissection, lung cancer surgery, eye surgery, dental surgery,...? "Surgery" needs to be defined and specified in this manuscript!
---

## [Author Response · Author response to Decision Letter 0]

29 Mar 2021

Dear Robert Jeenchen Chen, 

We are truly honored of the interest that the Editorial team of PLOS One has paid to our work.

We would like to thank the editor their thoughtful comments and constructive suggestions, which helped us to improve the quality of this report. Please find enclosed our revised manuscript and the rebuttal letter that specifically addresses every point they raised.

Sincerely yours,

Dr Michael Baboudjian 

COMMENTS FOR THE AUTHOR

Editor:

The scope of "surgery" is ill-defined. Did it include open heart surgery, brain surgery, and some other major surgery? Or just urology surgery? Please specify upfront in the Title. 

We thank the editor for this justified comment. This part has been completed in the Methods Section as follow:

“The patients were operated on in one of the hospital's seven surgical departments: urology, digestive, plastic, gynecological, otolaryngology, gynecology or maxillofacial surgery. »

Thus, cardiovascular, thoracic, orthopedic and brain surgeries were not included in this study.

In the Methods, the IRB is only approved in urology, not other surgical specialty. Was it legal to include non-urology surgery cases? 

The institutional review board approval from our hospital (Ap-Hm: Assistance Publique des Hôpitaux de Marseille) was provided for all surgical departments, not only for urology. In addition to the IRB, we have requested additional approval from the ethics committee of the French urology association. This is an ethics committee like any other, which can give its approval for a study that does not include only urology patients.

Please specify the scope of "surgery" in the Methods. Open heart surgery? TAVR? TEVAR? EVAR? Brain surgery, Whipple surgery, Bentall surgery, type-A aortic dissection, lung cancer surgery, eye surgery, dental surgery,...? "Surgery" needs to be defined and specified in this manuscript!

As previously described, this part has been completed in the Methods Section as follow:

“The patients were operated on in one of the hospital's seven surgical departments: urology, digestive, plastic, gynecological, otolaryngology, gynecology or maxillofacial surgery. »

---

## [Editor Report · Decision Letter 1]

1 Apr 2021

PONE-D-21-08635R1

Is Surgery Safe During the COVID-19 Pandemic? A Multi-Disciplinary Study

PLOS ONE

Dear Dr. Baboudjian,

Thank you for submitting your manuscript to PLOS ONE. After careful consideration, we feel that it has merit but does not fully meet PLOS ONE’s publication criteria as it currently stands. Therefore, we invite you to submit a revised version of the manuscript that addresses the points raised during the review process.

Please do not only reply but also revise the title, abstract, and main text accordingly.

We look forward to receiving your revised manuscript.

Kind regards,

Academic Editor

PLOS ONE

Journal Requirements:

Additional Editor Comments (if provided):

1. Not only in "reply", please also define and specify the specialties/scopes of "surgery" in "Title", "Abstract", and "Methods" in the main text. The revision fails to show it. For example, please add "minor elective surgery" or modify otherwise because it does not include cardiac, neuro-, emergent, or other major surgical cases.

2. In the revised Methods, you specified "all seven surgical departments...." What are these seven surgical departments are unknown to readers.

3. Your results and conclusion clearly do not apply to cardiac, vascular, orthopedic, or other major surgical cases (I myself is a cardiac surgeon so I care if your research applies to my specialty or not.). Please specify minor surgery upfront in Title and Abstract.

4. Did your cases include emergent or trauma cases?

5. Please revise well and do specify/define/limit your "surgery" which is different from the perception of "surgery" of many readers.

6. After clarification of "surgery", further review can be proceeded.

---

## [Author Response · Author response to Decision Letter 1]

8 Apr 2021

Dear Robert Jeenchen Chen, 

We are truly honored of the interest that the Editorial team of PLOS One has paid to our work.

Please find enclosed our revised manuscript and the rebuttal letter that specifically addresses every point they raised.

Sincerely yours,

Dr Michael Baboudjian 

 

COMMENTS FOR THE AUTHOR

1. Not only in "reply", please also define and specify the specialties/scopes of "surgery" in "Title", "Abstract", and "Methods" in the main text. The revision fails to show it. For example, please add "minor elective surgery" or modify otherwise because it does not include cardiac, neuro-, emergent, or other major surgical cases.

This comment is relevant. In fact, in our hospital there is no cardiac, vascular or neurological surgery. However, some of the patients included underwent major surgery such as cancer surgery (eg laryngectomy, cystectomy, gastrectomy) or kidney transplantation (n=15). In addition, as shown in Table 1, 28 included patients were acute surgical emergencies. Thus, all types of surgery were included: minor and major, elective and urgent surgery.

However, due to the lack of available resuscitation bed during the pandemic period, all surgeries were scheduled to not use postoperative resuscitation care and therefore, were mainly minor surgeries. As suggested by the editor, some changes have been added in the Title, the Abstract and The Manuscript:

Title: Is Minor Surgery Safe During the COVID-19 Pandemic? A Multi-Disciplinary Study

Abstract: Our study included all adult patients who underwent minor surgery in one of the seven surgical departments of our hospital: urology, digestive, plastic, gynecological, otolaryngology, gynecology or maxillofacial surgery.

Methods: The patients were operated on in one of the hospital's seven surgical departments: urology, digestive, plastic, gynecological, otolaryngology (two departments), gynecology or maxillofacial surgery. In our hospital, there was no cardiac, vascular, neurological or orthopedic surgery.

Discussion (Limitations): Finally, the present study reports on a very heterogeneous surgical population and covers a wide range of planned elective surgeries and a variety of emergency surgical presentations. However, our study population consisted mainly of patients who underwent minor elective surgery. This observation should be related to a significant decrease in acute surgical emergencies as observed in previous studies found in the literature [26]. In addition, major surgeries such as cardiac, vascular or neurological surgeries requiring postoperative resuscitation care were not included in our series.

2. In the revised Methods, you specified "all seven surgical departments...." What are these seven surgical departments are unknown to readers.

This comment is justified. As suggested by the reviewer, this data has been added in the Methods section: 

“All of the seven surgical departments in our hospital were involved in our study: urology, digestive, plastic, otolaryngology, gynecology or maxillofacial surgery.”

3. Your results and conclusion clearly do not apply to cardiac, vascular, orthopedic, or other major surgical cases (I myself is a cardiac surgeon so I care if your research applies to my specialty or not.). Please specify minor surgery upfront in Title and Abstract.

This is a justified comment. As suggested, we specified in the title and the abstract that the surgeries were mainly minor.

4. Did your cases include emergent or trauma cases?

Yes, some cases were acute surgical emergencies (n=28, 5.1% of all cases, Table 1). These patients therefore represent a very small part of our included population. This limitation is notified in the Discussion section:

“However, our study population consisted mainly of patients who underwent minor elective surgery. This observation should be related to a significant decrease in acute surgical emergencies as observed in previous studies found in the literature [26]. »

5. Please revise well and do specify/define/limit your "surgery" which is different from the perception of "surgery" of many readers.

As suggested by the reviewer, we have specified the type of surgery included throughout the manuscript: mainly minor and elective. In addition, we specified in the Methods part and in the Limits part that neither cardiac, vascular or neurological surgery was included in our study.

6. After clarification of "surgery", further review can be proceeded.

Dear editor, thank you for your comments which greatly improve the relevance and applicability of this manuscript.

---

## [Decision Letter · Decision Letter 2]

21 Apr 2021

Is Minor Surgery Safe During the COVID-19 Pandemic? A Multi-Disciplinary Study

PONE-D-21-08635R2

Dear Dr. Baboudjian,

We’re pleased to inform you that your manuscript has been judged scientifically suitable for publication and will be formally accepted for publication once it meets all outstanding technical requirements.

Kind regards,

Academic Editor

PLOS ONE

Additional Editor Comments (optional):

Reviewers' comments:

Reviewer's Responses to Questions

**Comments to the Author**

1. If the authors have adequately addressed your comments raised in a previous round of review and you feel that this manuscript is now acceptable for publication, you may indicate that here to bypass the “Comments to the Author” section, enter your conflict of interest statement in the “Confidential to Editor” section, and submit your "Accept" recommendation.

Reviewer #2: All comments have been addressed

Reviewer #3: All comments have been addressed

2. Is the manuscript technically sound, and do the data support the conclusions?

Reviewer #2: Yes

Reviewer #3: Yes

3. Has the statistical analysis been performed appropriately and rigorously? 

Reviewer #2: Yes

Reviewer #3: Yes

4. Have the authors made all data underlying the findings in their manuscript fully available?

Reviewer #2: Yes

Reviewer #3: Yes

5. Is the manuscript presented in an intelligible fashion and written in standard English?

Reviewer #2: Yes

Reviewer #3: Yes

6. Review Comments to the Author

Reviewer #2: This is an interesting study, and I approve of its publication.This study is a pioneer and I believe it will help in the future with COVID-19.

Reviewer #3: The authors' revised manuscript had specifically addressed every point raised by the reviewers with appropriate response.

7. PLOS authors have the option to publish the peer review history of their article (what does this mean?). If published, this will include your full peer review and any attached files.

Reviewer #2: No

Reviewer #3: **Yes: **Chao-Yang Chen

---

## [Editor Report · Acceptance letter]

29 Apr 2021

PONE-D-21-08635R2 

Is Minor Surgery Safe During the COVID-19 Pandemic? A Multi-Disciplinary Study 

Dear Dr. Baboudjian:

I'm pleased to inform you that your manuscript has been deemed suitable for publication in PLOS ONE. Congratulations! Your manuscript is now with our production department. 

Kind regards, 

on behalf of

Dr. Robert Jeenchen Chen 

Academic Editor

PLOS ONE